# Endocan as a Novel Biomarker for Endothelial Dysfunction and Cardiovascular Prognosis in ST-Elevation Myocardial Infarction: A Contemporary Literature Review

**DOI:** 10.3390/jpm16010007

**Published:** 2025-12-29

**Authors:** Sourabh Khatri, Pooja Suchday, Ananth Guddeti, Supritha Nanna, Shashank Gupta, Haritha Darapaneni, Adil Sarvar Mohammed, Rupak Desai, Hassaan Imtiaz

**Affiliations:** 1Department of Internal Medicine, Independence Health Westmoreland Hospital, Greensburg, PA 15601, USA; sourabh.khatri@independence.health; 2GCS Medical College, Hospital and Research Center, Ahmedabad 382345, India; phs15899@gmail.com; 3Department of Internal Medicine, SUNY Upstate Medical University, Syracuse, NY 13210, USA; guddetia@upstate.edu; 4Department of Family Medicine, Red Deer Regional Hospital Center, Red Deer, AB T4N 4E7, Canada; suprithananna@gmail.com; 5Department of Internal Medicine, West Anaheim Medical Center, Anaheim, CA 92804, USA; sgupta4@primehealthcare.com; 6Department of Internal Medicine, Central Michigan University College of Medicine, Saginaw, MI 48602, USA; harit1d@cmich.edu; 7Independent Researcher, Atlanta, GA 30033, USA; drrupakdesai@gmail.com; 8Department of Internal Medicine, McLaren Bay Region, Bay City, MI 48708, USA; hassaan.imtiaz@mclaren.org

**Keywords:** endothelial cell-specific molecular 1, endocan, ESM-1, biomarker, ST-elevated myocardial infarction (STEMI), endovascular dysfunction

## Abstract

The pathophysiology of ST-elevated myocardial infarction (STEMI) extends beyond coronary artery occlusion to include microvascular and endothelial dysfunction, both of which critically influence outcomes. Endocan, a soluble dermatan sulfate proteoglycan secreted by endothelial cells, has emerged as a novel biomarker of endothelial activation and dysfunction. Recent studies suggest that elevated endocan levels may carry prognostic significance in patients with STEMI, particularly those undergoing percutaneous coronary intervention (PCI). A comprehensive search of PubMed, Cochrane Library, and Google Scholar was conducted to identify studies evaluating endocan as a prognostic biomarker in STEMI. Review articles, case reports, case series, and experimental studies were excluded. Seven clinical studies, comprising sample sizes ranging from 80 to 320 patients, met the inclusion criteria. Across these studies, endocan levels were analyzed in relation to established prognostic markers and clinical outcomes. Key findings demonstrated that higher endocan levels correlated with stress hyperglycemia (r = 0.21, *p* < 0.05), higher SYNTAX scores, and worse in-hospital outcomes. A cutoff value of 1.7 ng/mL predicted STEMI with 76.1% sensitivity and 73.6% specificity. Elevated endocan levels also showed positive correlations with the TIMI risk score, major adverse cardiovascular events (MACE), and were identified as independent predictors of incomplete ST-segment resolution (STR) (*p* = 0.044) and no-reflow phenomenon (NRP) (*p* < 0.001, OR = 2.39, 95% CI = 1.37–4.15). Collectively, the evidence indicates that endocan is strongly associated with endothelial dysfunction, MACE, NRP post-PCI, and impaired reperfusion. Moreover, traditional prognostic indices such as TIMI and SYNTAX scores appear to correlate with circulating endocan levels. However, variability in reported cutoff values across studies highlights the need for larger, multicenter trials with standardized endpoints to establish endocan’s diagnostic and prognostic utility in STEMI.

## 1. Introduction

### 1.1. Endocan: Molecular Structure, Physiological Role, and Association in STEMI

Endocan, also known as endothelial cell-specific molecule-1 (ESM-1), is a 50 kD soluble dermatan sulphate produced and activated by the vascular endothelium. It is a novel biomarker of endothelial damage, inflammatory reactions in cardiovascular disease (CVD) patients. Endocan is a cell adhesion molecule and an indicator of angiogenesis, playing a role in facilitating VEGF factors and upregulating the inflammatory response. Therefore, it has the potential to regulate treatment strategies [1]. Additionally, studies have shown that high levels of endocan are associated with worse clinical outcomes in STEMI patients [2,3,4]. Furthermore, a cutoff value of an endocan level of 1.7 (ng/mL) suggests the occurrence of STEMI, with a sensitivity of 76.1% and 73.6% specificity, helping predict the outcome [5]. According to Chong-Rong Qiu, serum levels of ESM-1 were higher in type-2 diabetes mellitus (T2DM) patients with stress hyperglycemia in STEMI [6].

### 1.2. Reported Inconsistencies and Emerging Evidence

According to Hassanpour et al., higher endocan levels are associated with higher ApoB, and Apo-B/ApoA1 ratio can be considered as a diagnostic marker in monitoring patients with STEMI [1]. However, Chong-Rong Qiu et al., 2016, suggests that there is some correlation of high ESM-1 levels in STEMI patients with T2DM. Further studies are needed to understand the significance of endocan levels in patients with vascular disease in T2DM [6]. Furthermore, Chong-Rong Qiu et al., 2016, provided some evidence that higher ESM-1 levels > 1.01 ng/mL were considered to be an independent predictor of MACE [6]. However, additional research is required to evaluate the effects of treatment. An association was also found between ESM-1 levels and NRP levels in post-PCI patients that may help detect patients with unsuccessful reperfusion and worse clinical outcomes [2].

Studies have suggested that endocan (ESM-1) levels have significant association with cardiovascular disorders, particularly ACS with NSTEMI; however, all past studies had different measurement parameters, answering the same clinical question of associating endocan/ESM-1 with ACS/STEMI. Our review aims to combine all the past evidence through the literature and provide future direction for further large-scale study.

### 1.3. Epidemiology and Clinical Impact of Endocan over STEMI

Coronary artery disease (CAD) accounts for over 7.4 million deaths and cases of disability worldwide, mainly attributed to atherosclerosis. Atherosclerosis plays a significant role in the inflammatory process, causing endothelial dysfunction and leading to acute coronary syndrome (ACS), and STEMI is the main cause of acute myocardial infarction [1]. Cardiovascular disease is a common complication in patients with diabetes, and studies have shown an association between stress hyperglycemia and STEMI due to endothelial dysfunction [6]. According to a study, around 3–71% of patients with acute myocardial infarction (AMI) are affected by stress hyperglycemia [6].

ST-segment elevation myocardial infarction (STEMI) requires urgent attention to reperfusion in order to avoid complications as it can cause thrombosis in an epicardial coronary artery through atherosclerosis. Hence, primary percutaneous coronary intervention (PCI) has become the most effective treatment strategy for STEMI [4].

Although primary PCI restores coronary blood flow in most patients with TIMI-3, some still exhibit poor restoration, and in spite of vessel patency. Therefore, assessing the inflammatory markers such as endocan, which play an important role in endothelial function, remains clinically relevant [2].

### 1.4. Biomarkers in Cardiovascular Disease: Discussion of Traditional and Emerging Biomarkers in Acute Coronary Syndromes

In patients with STEMI, several traditional biomarkers such as Troponin, CRP, and copeptin have been used. Among these, troponins released from ischemic cardiomyocytes are considered a highly specific marker used in diagnosing myocardial infarction (MI). Similarly, CRP is used to predict the prognosis of MI and the risks of post-infarction complications [7]. Copeptin is a nonspecific biomarker used to predict all-cause mortality in heart failure patients with reduced ejection fraction, but it can also be elevated in other conditions [8].

Emerging biomarkers that require further studies include IL6, which is elevated in adverse cardiac events and ACS, Soluble CD40 Ligand, which although increased in ACS, remains debatable according to some studies. Galectin-3 (Gal-3) is associated with myocardial fibrosis and inflammation, and it plays a critical role in predicting outcome in the heart. IL-37, which is elevated not only in acute coronary syndrome but also in patients with adverse cardiac events, is another emerging biomarker. PAPP-A is another marker which can be sensitive, specific, and an early marker which can be used in the diagnosis of ACS [8]. However, the novel biomarker of our interest in this study is endocan (ESM-1), since according to current evidence under study, endocan levels have been shown to increase in patients with STEMI, carrying diagnostic value, and may predict the outcome and prognostic importance, represent endothelial dysfunction, and explain underlying mechanisms explaining MACEs and poor outcome including non-STR and NRP, despite vascular patency.

## 2. Pathophysiological Role of Endocan in STEMI

### 2.1. A. Endothelial Activation

The endothelium plays a crucial role in vascular homeostasis by regulating vascular tone, proliferation of vascular smooth muscle cells, immune cell adhesion, and inflammation. Endothelial dysfunction is defined as an alteration in its vasoprotective homeostatic function [9]. Acute-phase proteins, cytokines (including IL-6, IL-1, and tumor necrosis factor-alpha [TNF-α]), and adhesion molecules (intercellular adhesion molecule [ICAM]. and vascular cell adhesion molecule [VCAM]) serve as strong evidence-based indicators of endothelial dysfunction [10,11] (Figure 1). Studies have shown that endocan is significantly elevated in various inflammatory disorders with endothelial dysfunction, such as diabetes, stress hyperglycemia, and psoriasis [6,12,13,14].

Moreover, in patients with cardiovascular disease, endocan levels are significantly increased and are independently related to soluble intercellular adhesion molecule-1 (sICAM-1) and soluble vascular cell adhesion molecule-1 (sVCAM-1) levels [15,16], suggesting their role in regulating cell-to-cell adhesions, leukocyte adhesion, and migration—a process crucial to the development and progression of myocardial infarction (Figure 1).

Intermittent hypoxia significantly upregulates the expression of endocan through the hypoxia-inducible factor-1α (HIF-1α)/VEGF pathway, thereby enhancing the expression of ICAM-1 and VCAM-1 and promoting the adhesion between monocytes and endothelial cells [17]. Additionally, there is a close relationship between platelet activation and erosion of the glycocalyx. Fundamentally, platelets from patients with acute coronary syndromes release matrix metalloproteinase-2 (MMP-2), a major source of glycocalyceal damage [18], while, in turn, MMP-2 induces further platelet activation. Nonlaminar flow patterns also stimulate thioredoxin-interacting protein (TXNIP) expression and NLRP3-inflammasome activation [19]. This cascade promotes mitochondrial dysfunction and the resultant oxidative stress and impaired glucose utilization, leading to endothelial cell apoptosis [20].

Unlike most proteoglycans, which are mainly located in the extracellular matrix (ECM), endocan is mainly secreted in the blood upon glycocalyx destruction and is detectable in the serum [21].

Together, these findings suggest that endocan may have the potential to serve as a surrogate marker for endothelial dysfunction in addition to playing a functional role in endothelium-dependent pathological disorders [22].

### 2.2. B. Inflammation

The role of inflammation in the pathophysiology of ACS has been well established [23]. It induces structural remodeling of the coronary microvasculature with the help of mast cells, platelets, and neutrophil activation, leading to erosion of the endothelial glycocalyx—the endothelial barrier made up of various proteoglycans like syndecan, glypican, and endocan [24].

This enzymatic erosion of glycocalyx leads to the disruption of laminar blood flow, facilitates endothelial platelet interaction, and leads to impairment of platelet and vascular responsiveness to autocidal coronary vasodilators, such as nitric oxide, prostacyclin, and hydrogen sulfide, thus predisposing to coronary vasoconstriction and micro-thrombus formation [25].

STEMI is often precipitated by the rupture of thrombotic plaques. Following plaque rupture, neutrophils and monocytes are recruited to the site of injury, where they contribute to further inflammation and myocardial damage. The neutrophil-to-lymphocyte ratio (NLR) has been studied as a potential prognostic marker in STEMI, due to its reflection of systemic inflammation [26]. Inflammatory markers IL-1β and IL-6 have also been associated with increased thrombolysis in myocardial infarction (TIMI) risk score in STEMI, highlighting their role in risk stratification [23]. Additionally, IL-6 and TNF-α have been correlated to atherosclerotic plaque destabilization [27]. So, how does this help us bridge the gap between the relation of endocannabinoids to inflammation and further to STEMI?

Pro-inflammatory factors such as TNF-α and IL-6 induce the expression of endocan genes and show a positive correlation, whereas the anti-inflammatory factor IL-10 is negatively correlated with endocan. IFN-γ has no impact on the expression of endocan genes induced by TNF-α [14,28].

Endocan elicits severe vascular inflammatory responses in cellular and animal experimental models of sepsis [29]. In human umbilical venous cells, it has been shown to increase the release of chemokines interleukin-8 and monocyte chemotactic protein-I [MCP-I], induce the expression of ICAM-1, VCAM-1, and E-selectin, and stimulate the mitogen-activated protein kinase protein (MAPK) signaling pathway and NF-*κ*B (nuclear factor kappa B). Additionally, endocan in high serum concentrations enhances leukocyte migration and induces endothelial cytoskeletal rearrangement, leading to cellular contraction and alteration of cellular permeability [30].

Endothelial damage increases the production of reactive oxygen species (ROS), which can reduce the production of NO [31]. The increased production of ROS and the impairment of NO availability can induce and maintain the inflammatory state of the blood vessel wall [28]. Furthermore, inflammation-induced microvascular dysfunction can complicate reperfusion efforts and exacerbate the no-reflow phenomenon.

### 2.3. C. Angiogenesis

Endocan plays a role in neo-angiogenesis via its interaction with vascular endothelial growth factor (VEGF), as suggested by Ozaki, K. et al. in a study that correlates elevated serum endocan levels with greater number of tumors, and more severe vascular invasion in patients with hepatocellular carcinoma [21].

Endocan enhances endothelial cell proliferation and migration, which are essential for new blood vessel formation during tissue repair following ischemic events [28]. Furthermore, the pro-inflammatory state it creates might also play a significant role in recruiting additional growth factors needed for angiogenesis at the site of myocardial ischemia.

### 2.4. D. Interplay with Ischemia-Reperfusion Injury

Endocan plays a significant role in modulating microvascular dysfunction and tissue reperfusion, particularly in the context of STEMI (Figure 2). Endothelial dysfunction, characterized by increased permeability, leukocyte adhesion, and inflammatory mediator release [32], contributes to microvascular obstruction. This increases myocardial oxygen demand and the chances of reperfusion injury.

Post-PCI, some patients might exhibit overt impairment of myocardial reperfusion despite successful opening of the occluded vessel. The blood flow in these patients cannot reach TIMI 3 after post-PCI, which is called the “no-reflow phenomenon (NRP)” [33]. The NRP might be related to pathological factors such as endothelial dysfunction, oxidative stress, ischemic and reperfusion injury, platelet aggregation, thrombus formation, distal thromboembolization, coronary vasomotor dysfunction, and individual susceptibility, and is considered to be an inflammatory state [2].

The mortality rate in STEMI and the risk of congestive heart failure, malignant arrhythmia, and sudden cardiac death significantly increase in the patients with the NRP post-PCI. Moreover, a high endocan level on hospital admission might serve as an independent predictor of worse cardiovascular outcomes and higher TIMI risk scores in patients with ACS [3].

On the other hand, during reperfusion following MI, endocan may facilitate the restoration of blood flow by promoting angiogenesis post the ischemic event, a process critical for tissue recovery.

As endocan is a marker of inflammation and also plays a significant role in neo-angiogenesis, it might reflect a delicate balance between the detrimental inflammation and the beneficial angiogenesis that compensates for the ischemic event.

### 2.5. E. Comparison to Other Biomarkers: Troponin, CRP, and Natriuretic Peptides

In patients with acute coronary syndromes (ACS), troponin I (TnI), C-reactive protein (CRP), and B-type natriuretic peptide (BNP) each predict adverse cardiac events (Table 1). Troponin is the gold standard for detecting myocardial injury, CRP provides insights into systemic inflammation, and endocan reflects endothelial dysfunction, inflammation, reperfusion defects, and angiogenesis, and natriuretic peptides assess cardiac stress and the risk of heart failure. A multi-marker strategy might help with more accurate risk stratification, patient monitoring, and individualization of treatment.

## 3. Clinical Evidence Linking Endocan to STEMI Outcomes

Current clinical evidence suggests endocan as a biomarker for endothelial dysfunction and ischemic conditions along with a positive correlation with other biomarkers like apolipoprotein-B (APO-B) [1]. Endocan, along with galectin-3, can predict a higher risk of insufficient myocardial reperfusion and worse clinical outcomes after post-PCI [4,38]. Studies also suggested a positive and independent correlation of endocan levels with MACE, indicating a poor prognosis with elevated levels [3,5].

“No reflow phenomenon (NRP)” after post-PCI is an important predictor and risk factor for worse clinical outcomes after STEMI and PCI associated with MACE, heart failure, arrhythmia, and risk of sudden cardiac death. As discussed in the pathophysiology section, the NRP is associated with endothelial dysfunction, and endocan is a representative biomarker for endothelial dysfunction. Also, endocan has been widely suggested as a biomarker indicating decreased myocardial perfusion and the NRP [2,4].

Endothelial dysfunction in other conditions, including diabetes mellitus, has been noted to have elevated endocan levels along with vascular disorders like STEMI. However, higher levels of endocan or significant differences in endocan levels have been observed in acute vascular endothelial disorders like STEMI as compared to conditions that are chronic, like diabetes, which suggests a cutoff value for endocan levels can help in diagnosing/prognosing patients with acute presentations with STEMI [12]. Studies suggest endocan can represent endothelial dysfunction and a risk for cardiovascular disorders in prospective follow-up [39]. Also, elevated endocan levels can represent a worse prognosis for chronic stable heart failure-related mortality and hospitalizations [40].

Kundi et al. suggested a direct correlation of endocan with the SYNTAX score, which is a semi-quantitative measurement of the complexity of coronary artery disease, and a value of endocan > 1.7 ng/mL suggested the presence of STEMI with a sensitivity of 76.1% and specificity of 73.6% [5], and an endocan value of >2.7 ng/mL has 89.6% sensitivity and 74.2% specificity for the prediction of the NRP [2].

The current evidence in the literature suggests endocan as a promising biomarker for endothelial dysfunction and the prognosis associated with STEMI, including MACE and the NRP. However, the studies do not represent a long-term prognosis and follow with endocan levels. Most studies are either cross-sectional or have minimal follow-up (<6 months). Also, most studies are either a single-center trial or multicenter with a sample size of *n* < 500. Future research should be focused on multicenter studies with representative populations along with stronger and longer clinical follow-up.

## 4. Methodological Considerations in Endocan Research

### 4.1. A. Assay Variability and Standardization

Endocan, a soluble dermatan sulfate proteoglycan (DSPG) secreted by endothelial cells, is typically measured in blood due to the non-invasive nature of sampling. Among available techniques—such as immunohistochemistry, electrophoretic, and various immunoassays—the ELISA sandwich assay remains the most widely used and validated method, offering high sensitivity and specificity.

In the included studies, endocan levels were uniformly measured using ELISA sandwich kits, ensuring methodological consistency. Blood samples were centrifuged at 3200 rpm for 10 min at 4 °C to separate serum, which was stored at −80 °C until analysis. The detection limit of standard ELISA kits is approximately 0.02 ng/mL, and measurements were performed in duplicate to enhance reliability.

### 4.2. B. Study Design and Limitations

Our literature search included a meticulous search through PubMed, Scopus, Embase, Cochrane Registry, and Google scholar research databases. The studies included were recent to 1 January 2015 and searched through end of 28 February 2025. The keywords were included “Endocan,” “ESM-1,” “ST-elevated myocardial infarction (STEMI),” “acute coronary syndrome (ACS),” “reperfusion injury,” “TIMI score,” “SYNTAX score,” “angiogenesis,” and “endothelial dysfunction.”

Inclusion criteria included were observational/cross-sectional studies measuring patients admitted to hospital with ACS and STEMI along with measuring endocan levels and correlated prognostic patient outcomes. Only studies with a sample size at least >10 and published after 1 January 2015 were included.

Exclusion criteria were studies with sample size < 10, studies older than 1 January 2015. Through the database, review articles, case reports, case series, letters to editors, preprints, clinical trials, and experimental studies, if found, were also excluded.

After a careful manual review of two researchers independently working (S.K. and P.S.), an initial 10 articles were screened-in and then 3 articles were screened out following the predefined search criteria and quality assessment methods. Disagreements between these 2 authors were resolved by consensus, including a third author (R.D.). The exact terms applied in each database from this search strategy are shown in the Prisma diagram as below (Figure 3).

After a meticulous review, a total of seven studies met inclusion criteria with a combined sample size of 1011 patients (ranging from 80 to 320 per study). It is to be noted that all studies included did not have a uniform outcome/prognosis assessment. All studies evaluated ACS patients with STEMI undergoing PCI, assessing correlations between endocan levels and outcomes such as no-reflow phenomenon, TIMI score, hs-CRP, SYNTAX score, type 2 diabetes, and incomplete-ST resolution.

Most studies utilized control cohorts from the same population, ensuring comparability and strengthening internal validity. Uniform data collection across single-center cohorts further enhanced methodological consistency. However, potential confounders such as smoking, alcohol use, hypercholesterolemia, obesity, and postmenopausal status may influence endocan levels independently, limiting external generalizability. Nonetheless, the findings support endocan’s potential as a biomarker for both angiogenesis and prognostic assessment following STEMI treated with PCI.

### 4.3. C. Endocan as a Prognostic Tool in Patients with STEMI

Endocan is a novel biomarker that can potentially be used to either predict the severity of STEMI during a hospital admission to the ED. In the study by Ziaee et al. [3], endocan levels on admission into the ED were taken, and there was a significant positive linear association between endocan and the TIMI score in the STEMI subset, suggesting the potential use of endocan as a prognostic factor for ACS patients upon admission to the ED. Another study by Kundi et al. [5] suggested that high serum endocan levels were associated with high in-hospital mortality and more severe coronary atherosclerosis. The syntax score, which is calculated during a coronary angiography procedure by cardiologists, has been historically found to be linearly associated with worse prognosis, suggesting a complex condition. Higher syntax scores were associated with higher endocan levels in this study, which further supports our discussion regarding endocan to be used as a biomarker to predict the severity of STEMI upon admission. Moreover, a cutoff endocan level of 1.7 (ng/mL) predicted the presence of STEMI with a sensitivity of 76.1% and specificity of 73.6%.

Endocan can also be used to predict reperfusion and as a prognostic tool after PCI. As mentioned in Dogdus et al. [2], endocan levels were significantly higher in the NRP (+) group, identified as an independent predictor (OR = 2.39) with a cutoff value > 2.7 ng/mL (89.6% sensitivity, 74.2% specificity), suggesting that endocan can also be used as a predictive factor for post-PCI. Additionally, as per the study by Turan et al. [4], there was also a significant correlation between endocan levels and ST segment resolution (STR) following PCI. Their hypothesis was that elevated levels of endocan, as an indicator of endothelial dysfunction and inflammation, may be surrogate markers of insufficient microvascular perfusion at the myocyte level post-PCI, which might contribute to an incomplete STR. This study also suggested the possibility of potential lack of reperfusion causing an increase in endocan levels, further supporting the study by Dogdus et al. [2]. Additionally, in the study by Qui et al. [6], a multiple factors logistic regression analysis in the study indicated that an ESM-1 level > 1.01 ng/mL was an independent predictor of MACE. This further supports that endocan levels can be used as a predictor to understand MACE (Table 2).

### 4.4. D. Endocan and Apolipoprotein Association

It has been previously well documented that the ApoB, ApoA1, and APO-B/APO-A1 ratio can be considered a strong biomarker in the case of CVD, including STEMI patients. Interestingly, endocan levels have been correlated to apolipoproteins in STEMI patients [1]. Endocan significantly correlated to APO-B by the Pearson correlation coefficient in this study. It was concluded that the higher levels of endocan accompanied by a higher APO-B/APO-A1 ratio under ischemic conditions would be touted as diagnostic biomarkers for monitoring patients with STEMI. This further supports that endocan can be used as a prognostic factor in STEMI as it correlates with the Apo-B levels independently.

### 4.5. E. Endocan Levels in Patients with T2DM Following STEMI

Endocan levels can vary significantly in patients with T2DM and STEMI and T2DM patients without STEMI, as shown in the study by Qui et al. [6], which showed a difference of (1.25 ± 0.50 vs. 1.09 ± 0.16 ng/mL, *p* < 0.01). This suggests that endocan levels are further amplified by a STEMI event independently and eliminates the confounding nature of T2DM in patients with STEMI and T2DM. ESM-1 was independently increased in comparison to the control group with patients having T2DM as well (1.03 ± 0.03 vs. 1.17 ± 0.38 ng/mL, *p* < 0.03), which attributes to the independent nature of increased ESM-1 in diabetes [41], which has been well studied previously. Further studies on the additional burden and impact of STEMI in addition to T2DM need to be conducted to better understand the significance of STEMI in patients who are already diagnosed with T2DM.

### 4.6. F. Endocan Levels in Stress Hyperglycemia Following STEMI

Stress hyperglycemia after acute myocardial infarction was first reported in 1931; stress hyperglycemia in a setting of acute myocardial infarction increases all-cause mortality and MACEs [42,43]. In the study by Qui et al. [6], there was a significant increase in ESM-1 level in patients with stress hyperglycemia having STEMI compared with controls who did not have stress hyperglycemia post-STEMI. This could be explained by mitochondrial dysfunction in endothelial cells resulting in endothelial dysfunction and the inflammatory response to stress hyperglycemia, which increases inflammatory cytokines such as TNF Alpha and Interleukin 1-Beta, which in turn increases endocan levels. Endocan levels further correlating to stress hyperglycemia may play a crucial role in predicting MACEs in future studies.

## 5. Summary and Clinical Implications

Endocan levels have the potential to be a good screening tool for STEMI, as suggested by the linear correlation between endocan and TIMI score. It can further be incorporated in routine workup for ACS as an inflammatory marker; however, its clinical utility may be limited by confounding inflammatory conditions such as sepsis [44]. Further research to mark a cutoff endocan level in ACS and, more specifically, STEMI can be useful. ESM-1 levels can also be used for understanding the outcomes post-PCI, including MACE as suggested by Qui et al. [6]; however, the sample population is the study’s limitation, and further research with a bigger sample size into understanding the long-term outcomes in patients with elevated ESM-1 can be useful. It would be useful to see how the endocan levels play out throughout the course before STEMI and the time of intervention, focusing on survival, mortality, and long-term outcomes, including MACE, quality of life, and recurrent admissions.

## 6. Discussion and Research Gap

A better understanding of endocan with respect to other inflammatory conditions like sepsis, CKD, IBD, and tumor progression can help us narrow down endocan’s relationship to cardiovascular diseases; however, these associations remain exploratory and should be interpreted cautiously. It is a novel biomarker that has the potential to be used to further understand the long-term prognosis of ACS patients as well as understand the severity of the disease upon admission. However, future prospective studies with larger sample sizes are required to clarify its prognostic role.

Our study delineates a concise analytical review of the present evidence available on the research literature to establish endocan levels as a prognostic indicator particularly for patients with acute conditions like STEMI, undergoing PCI. In an era of endovascular coronary interventions, it is necessary to establish prognosis for endovascular dysfunction prior to every PCI, since it is important to explore why all endovascular coronary interventions vary in prognosis. This is especially important given present prognostic indicators do not account for post-PCI endovascular dysfunction which denote important aspects of post-procedure prognosis even when anatomical obstruction is corrected. Often, patients presenting to the Cath lab with acute condition like STEMI and needing coronary interventions have multiple comorbidities like uncontrolled diabetes/uncontrolled hyperglycemia, deranged kidney function, and chronic inflammatory disorders like chronic infections, etc. Post-procedure outcome and recurrent MACE events are still common in these patients, hinting dynamic flow obstruction with pathophysiological microvascular dysfunction.

Hassanpour et al. [1] suggested endocan levels correlated with Apo-B, which itself is an established marker of endothelial dysfunction. However, endocan levels are more specific to ischemic and acute levels of microvascular dysfunction. Similarly, galectin, which is similar to endocan levels, is a marker of endovascular dysfunction which needs to be studied more. Adding to the evidence, physiological phenomenon which predict endothelial and microvascular dysfunction post-PCI like no-reflow phenomenon and incomplete STR correlate to endocan levels.

However, an important limitation of our study is the heterogeneity of the outcome variables in the noted seven studies. The variable and heterogenous primary and secondary endpoints measurements in our included studies included MACE events, physiological phenomenon like NRT or incomplete STR, markers like Apo-B and Apo-A levels, and hs-CRP. Heterogenous variables measured could not be combined to have statistically relevant evidence that could not be measured and evidence cannot have a generalizability. Also of note, different studies denote different cutoff points for endocan levels, constraining to have particular value with established relevance. Additionally, variations in study design, follow-up duration, endpoints, and measurement techniques represent potential sources of bias and heterogeneity, which should be considered when interpreting these findings.

Further research highlighting Endocan’s correlation and establishing numerical cutoff value for prognosis is needed with bigger sample sizes. Future research can also help us incorporate Apo-B levels along with endocan into understanding the severity of cardiovascular disease. High cholesterol, alcohol, obesity, and smoking can also increase endocan levels, and future studies with stratification can help us remove the confounders and truly understand the nature of endocan levels in ACS patients. Stratification would also help us understand how much cholesterol, obesity, alcohol, and smoking really affect the endocan levels and provide further insight into those conditions as well and their relationship with endocan. This could help us establish a cutoff value of endocan for each inflammatory condition. This can help us specifically target ACS and the relation of ESM-1 levels to STEMI patients. It would also be interesting to understand how endocan levels vary in other ACS conditions such as NSTEMI and unstable angina. In addition, high-risk ECG phenotypes such as Wellens syndrome have been associated with critical coronary pathology and microvascular dysfunction despite transient symptoms. Case-based reports suggest that endothelial injury and altered coronary flow dynamics may contribute to this presentation. Given endocan’s association with endothelial activation, its potential role in refined risk stratification in such phenotypes warrants further investigation [45]. Understanding its role in angiogenesis can help us understand further how the myocyte repair correlates to prognosis and long-term outcomes. There is conflicting data as angiogenesis and myocyte repair are conducted by endocan at the same time it positively correlates to MACEs post-PCI. Further understanding this aspect can help us understand the nature of endocan and its balance between favorable outcomes and adverse events post- PCI.

## 7. Conclusions

Endocan is a promising novel biomarker for predicting endothelial dysfunction. In the current era of innovations and progression of understanding of the array of coronary artery disease, understanding the physiological and dynamic pathological mechanisms of coronary artery lesions is as important as understanding anatomic obstruction. While elevated endocan levels do represent escalating inflammation, it was observed that endocan levels were elevated significantly in STEMI as compared to chronic vascular inflammation like hyperglycemia and diabetes. One study reported no significant differences between NSTEMI and STEMI endocan levels [46]. However, one study reported a significant association of endocan levels with non-resolution of ST segment elevations post-PCI. Our current literature review suggests endocan is correlated with prognosticating scores like TIMI risk score, syntax scores, and MACEs, and in fact, physiological obstruction indicates endovascular dysfunction. As per our review, the diagnostic ability of endocan levels is still questioned; however, there is significant evidence beyond just mere consideration that endocan carries significant value in the prognostication of ACS patients with STEMI and can predict worse prognosis in terms of a variety of endpoints, including MACEs, NRP post-PCI, and non-STR. Future studies should be conducted with long-term follow-up to establish a diagnostic and prognostic application and cutoff value for endocan in ACS patients with STEMI.

## Figures and Tables

**Figure 1 jpm-16-00007-f001:**
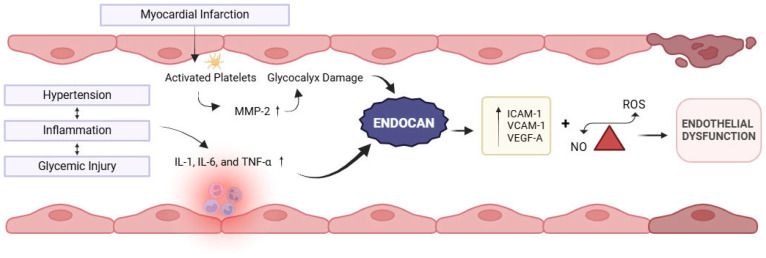
Representation of myocardial infarction-related activation of inflammatory cytokines and association of endocan with endothelial dysfunction. *Created in BioRender. Khatri, S. (2025) https://BioRender.com/38oz97b (accessed on 21 December 2025).*

**Figure 2 jpm-16-00007-f002:**
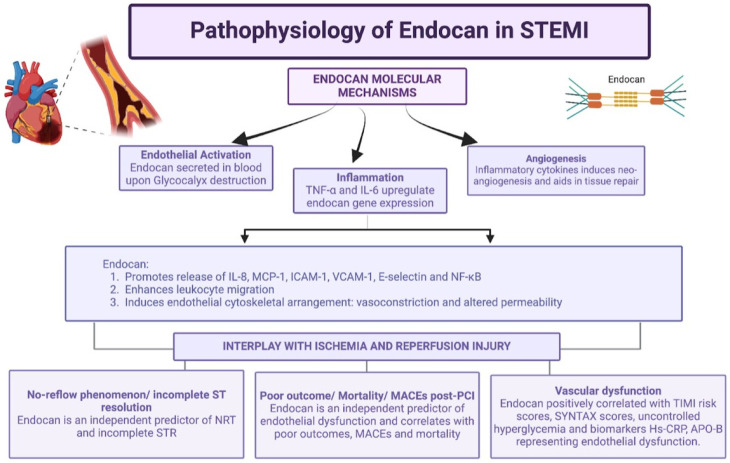
Representation of pathophysiology of endocan and interplay with ischemia and reperfusion injury. *Created in BioRender. Khatri, S. (2025) https://BioRender.com/pc58rkz (accessed on 21 December 2025).*

**Figure 3 jpm-16-00007-f003:**
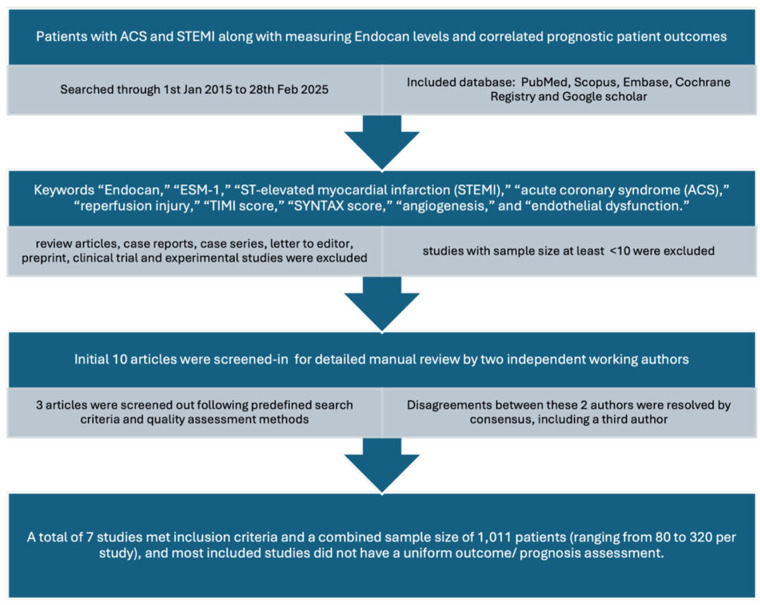
PRISMA-style flow diagram illustrating the literature search strategy, study screening, eligibility assessment, and final inclusion of studies evaluating endocan in STEMI.

**Table 1 jpm-16-00007-t001:** Important biomarkers with associated pathophysiology and role in STEMI outcomes.

Biomarkers	Pathophysiology	Role in STEMI
**Troponin**	Released from damaged cardiomyocytes due to ischemia with or without necrosis [8]. Amount correlates with the extent of myocardial damage	Highly sensitive and specific (gold standard) biomarker for diagnosing MI and independently predicts adverse cardiac outcomes [7,8].
**CRP**	Marker reflecting systemic inflammation produced in the liver.	Can predict prognosis of MI and the risk of post-infarction heart failure [7,34].
**Natriuretic Peptides (ANP, BNP, NT pro-BNP)**	Released in response to increased cardiac wall stress in heart failure. Serve as counterregulatory hormones for volume and pressure overload [35].	BNP level in the bloodstream has a predictive role for cardiovascular risk in the general population [8]. Elevated BNP and NT pro-BNP levels correlate to higher mortality (seven-fold higher with BNP level > 80 pg/mL) [36], and heart failure risk post-MI [37].
**Endocan**	Reflects endothelial dysfunction and inflammation biomarker.	High endocan level on hospital admission might serve as an independent predictor of worse cardiovascular outcomes and higher TIMI risk score in patients with ACS [3]. Elevated endocan in STEMI patients with stress hyperglycemia is associated with a higher risk of MACEs [12]. Endocan levels may help identify patients at risk of poor myocardial perfusion and the “NRP”, and adverse outcomes after PCI [2,4].

**Table 2 jpm-16-00007-t002:** Studies with clinical evidence of endocan association with STEMI outcome.

	Study	Sample Size	Study Type	Objective	Key Methods/Measurements	Follow-Up Duration	Primary Endpoint	Independent Prognostic Value of Endocan	Key Findings	Conclusion
1	Hassanpour et al., 2024 [1]	80 men	Cross-sectional study	Investigate serum endocan levels in STEMI patients and their correlation with apolipoproteins (APO-A1 and APO-B).	Measurement of serum endocan, APO-A1, and APO-B; correlation analysis between biomarkers.	Not applicable	Biomarker correlation [endocan vs. APO-A1, APO-B, APO-B/APO-A1 ratio.	Not assessed [no clinical outcomes or multivariable outcome model].	STEMI cases showed significant differences in APO-A1, APO-B, endocan, and the APO-B/APO-A1 ratio compared to controls; a significant positive correlation was found between endocan and APO-B.	High endocan level is an independent indicator of endothelial dysfunction and ischemic conditions, potentially related to APO-B.
2	Dogdus et al., 2021 [2]	137 STEMI patients	Prospective observational study	Evaluate whether serum endocan levels can predict the angiographic no-reflow phenomenon (NRP) in STEMI patients undergoing primary PCI.	Measurement of serum endocan; clinical and angiographic assessment; multivariate logistic regression; ROC curve analysis.	In-hospital	Angiographic no-reflow phenomenon (NRP).	Yes—independent predictor in multivariable logistic regression [OR 2.39, 95% CI 1.37–4.15].	Endocan levels were significantly higher in the NRP (+) group; identified as an independent predictor (OR = 2.39) with a cutoff value > 2.7 ng/mL (89.6% sensitivity, 74.2% specificity).	Endocan levels may help identify patients at higher risk for insufficient myocardial perfusion and worse outcomes post-PCI.
3	Turan et al., 2020 [4]	98 STEMI patients	Cross-sectional study	Assess the relationship of endocan and galectin-3 levels with ST-segment resolution (STR) in STEMI patients undergoing PCI.	Measurement of serum endocan and galectin-3; recording of SYNTAX scores; comparison between complete (≥70%) and incomplete (<70%) STR groups; regression analysis.	In-hospital	Incomplete ST-segment resolution [<70%].	Yes—independently predicted incomplete STR after adjustment.	Patients with incomplete STR had significantly higher endocan, galectin-3, and SYNTAX scores, along with adverse metabolic profiles and lower ejection fraction; both biomarkers independently predicted incomplete STR.	Elevated endocan (and galectin-3) levels may help identify patients at risk of poor myocardial perfusion and adverse outcomes after PCI.
4	Ziaee et al., 2019 [3]	320 ACS patients (STEMI, NSTEMI, or UA)	Prospective cross-sectional study	Evaluate the prognostic value of serum endocan in relation to the TIMI risk score and its association with major adverse cardiac events (MACEs) in ACS patients.	Measurement of serum endocan on admission; correlation with TIMI risk score and clinical outcomes; multivariate logistic regression analysis; determination of optimal cutoff values.	In-hospital + 30-day follow-up	MACEs (in-hospital death, HF, recurrent MI) and correlation with TIMI score.	Yes—independently associated with MACE in multivariable model.	A significant positive correlation was found between endocan levels, TIMI risk score, and MACE; optimal cutoffs varied by ACS subtype; endocan was independently associated with MACE.	A high endocan score was a predictor of MACE and a positive correlation with that of TIMI score in ACS patients.
5	Kundi et al., 2017 [5]	133 patients (88 STEMI vs. 45 normal coronary arteries)	Cross-sectional study	Determine whether admission endocan level can predict in-hospital mortality and correlate with coronary severity (SYNTAX score) in STEMI patients.	Measurement of serum endocan, hsCRP, peak troponin I, left ventricular ejection fraction, and SYNTAX score; ROC analysis to identify an optimal endocan cutoff.	In-hospital	Presence of STEMI and coronary severity (SYNTAX).	Yes—independently associated with STEMI presence [not long term outcomes].	Endocan independently correlated with STEMI presence and coronary severity; a cutoff of 1.7 ng/mL predicted STEMI with 76.1% sensitivity and 73.6% specificity; significant correlations with hs-CRP and SYNTAX score were noted.	High admission endocan level is an independent predictor of adverse cardiovascular outcomes and a higher coronary severity index in STEMI patients.
6	Qiu et al., 2017 [12]	72 T2DM patients with STEMI and 33 control subjects (total *n* = 105)	Pilot observational study	Analyze serum endocan (ESM-1) levels in T2DM patients with STEMI and assess correlations with inflammatory markers.	Measurement of serum ESM-1; comparisons between T2DM with vascular disease, without vascular disease, and controls; correlation analysis with hs-CRP and neutrophil-to-lymphocyte ratio.	Not applicable	Comparison of endocan levels with groups (T2DM STEMI vs. others).	Not assessed [no outcome-based multivariable model].	T2DM patients with STEMI had significantly higher ESM-1 levels compared to controls and newly diagnosed T2DM without vascular disease; positive correlations with hs-CRP and neutrophil-to-lymphocyte ratio were observed.	Serum ESM-1 may serve as a novel biomarker of endothelial dysfunction and is associated with vascular disease in T2DM patients.
7	Qiu et al., 2016 [6]	105 STEMI patients with stress hyperglycemia and 33 controls (total *n* = 138)	Pilot observational study	Assess the relationship between endocan (ESM-1) levels and stress hyperglycemia in STEMI patients, and evaluate its predictive value for MACEs.	Measurement of serum ESM-1; correlation with admission glucose levels; multivariate logistic regression to evaluate prediction of MACEs over a 3-month follow-up.	3 months	MACEs at 3 months.	Yes—independently predicted MACE [OR 3.01; 95% CI 1.05–8.64].	Patients with stress hyperglycemia had significantly higher ESM-1 levels; a positive correlation between ESM-1 and glucose levels was noted; ESM-1 levels > 1.01 ng/mL independently predicted MACEs (OR = 3.01).	Elevated ESM-1 in STEMI patients with stress hyperglycemia is associated with a higher risk of MACEs, supporting its role as an independent prognostic biomarker.

## Data Availability

No new data were created or analyzed in this study. Data sharing is not applicable to this article.

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
