# Peer review of "Endocan as a Novel Biomarker for Endothelial Dysfunction and Cardiovascular Prognosis in ST-Elevation Myocardial Infarction: A Contemporary Literature Review"

_jpm, 2025, doi:10.3390/jpm16010007_

Round 1
Reviewer 1 Report
Comments and Suggestions for Authors
Thanks to authors for their contribution. The manuscript offers a timely and generally well-structured review of endocan as a marker of endothelial dysfunction and prognosis in ST-elevation myocardial infarction (STEMI), integrating pathophysiological context with clinical data. The topic holds clear relevance for clinicians and researchers addressing acute coronary syndromes and primary percutaneous coronary intervention (PCI).
To strengthen the manuscript, it is important to provide a clearer description of the review methodology. Please specify, in greater detail, the databases consulted, the time frame covered, the full search strings, and the inclusion and exclusion criteria employed, as well as the numbers of records identified, screened, and included. A concise PRISMA-style flow diagram and a summary table of the included studies would substantially enhance transparency and readability.
Additionally, the pronounced heterogeneity of the available studies, encompassing study design, sample size, timing of endocan measurement, cut-off values, assay methods, and adjustment for confounders, should be discussed more explicitly. This discussion should address the implications of these limitations for the strength and generalizability of the conclusions.
It would also be beneficial to more clearly delineate what is novel in this review relative to previous work on endocan in cardiovascular disease. In particular, emphasize the specific focus on STEMI treated with PCI, microvascular dysfunction and no-reflow, and the interactions with diabetes or stress hyperglycemia. Some sections, especially the pathophysiological discussion and the comparison with other biomarkers, could be tightened to foreground the mechanisms and aspects most relevant to STEMI and to highlight the potential incremental value of endocan beyond established markers.
Finally, the manuscript would benefit from careful english language editing to correct minor errors, ensure that all abbreviations are defined on first use, standardize terminology (notably regarding STEMI wording), and harmonize reference formatting. Overall, this is a relevant and promising review. Addressing these points would enhance its clarity, rigor, and impact.
Author Response
Reviewer 1:
Comments and Suggestions for Authors
1. Thanks to authors for their contribution. The manuscript offers a timely and generally well-structured review of endocan as a marker of endothelial dysfunction and prognosis in ST-elevation myocardial infarction (STEMI), integrating pathophysiological context with clinical data. The topic holds clear relevance for clinicians and researchers addressing acute coronary syndromes and primary percutaneous coronary intervention (PCI).
2. To strengthen the manuscript, it is important to provide a clearer description of the review methodology. Please specify, in greater detail, the databases consulted, the time frame covered, the full search strings, and the inclusion and exclusion criteria employed, as well as the numbers of records identified, screened, and included. A concise PRISMA-style flow diagram and a summary table of the included studies would substantially enhance transparency and readability.
Answer: Thank you for the expert suggestion, as per suggestion, we included details about the research databases used and the timeline for search, full search strings, inclusion and exclusion criteria, records identified, screened and finally included in the study- in a concise PRISMA-style flow diagram and seven studies included for analysis as in the summary Table 2. (The following changes have been incorporated in the edited manuscript on page 8-9, from lines 281-322)
4.2. B. Study Design and Limitations
Our literature search included a meticulous search through PubMed, Scopus, Embase, Cochrane Registry and Google scholar research databases. The studies included were recent to 1st January 2015 and searched through end of 28th February 2025. The keywords were included “Endocan,” “ESM-1,” “ST-elevated myocardial infarction (STEMI),” “acute coronary syndrome (ACS),” “reperfusion injury,” “TIMI score,” “SYNTAX score,” “angiogenesis,” and “endothelial dysfunction.”
Inclusion criteria included were observational / cross-sectional studies measuring patients admitted to hospital with ACS and STEMI along with measuring Endocan levels and correlated prognostic patient outcomes. Only studies with sample size at least >10 and published after 1st January 2015 were included.
Exclusion criteria were studies with sample size<10, studies older than 1st January 2015. Through the database, review articles, case reports, case series, letter to editor, preprint, clinical trial and experimental studies, if found were also excluded.
After a careful manual review of two researchers independently working (S.K. and P.S.), initial 10 articles were screened-in and then 3 articles were screened out following predefined search criteria and quality assessment methods. Disagreements between these 2 authors were resolved by consensus, including a third author (R.D.). The exact terms applied in each database from this search strategy are shown in the Prisma diagram as below.
After a meticulous review, a total of seven studies met inclusion criteria and a combined sample size of 1,011 patients (ranging from 80 to 320 per study). It is to be noted that, all studies included did not have a uniform outcome/ prognosis assessment. All studies evaluated ACS patients with STEMI undergoing PCI, assessing correlations between endocan levels and outcomes such as no-reflow phenomenon, TIMI score, hs-CRP, SYNTAX score, type 2 diabetes, and incomplete-ST resolution.
Most studies utilized control cohorts from the same population, ensuring comparability and strengthening internal validity. Uniform data collection across single-center cohorts further enhanced methodological consistency. However, potential confounders such as smoking, alcohol use, hypercholesterolemia, obesity, and postmenopausal status may influence endocan levels independently, limiting external generalizability. Nonetheless, the findings support endocan’s potential as a biomarker for both angiogenesis and prognostic assessment following STEMI treated with PCI.
Additionally, the pronounced heterogeneity of the available studies, encompassing study design, sample size, timing of endocan measurement, cut-off values, assay methods, and adjustment for confounders, should be discussed more explicitly. This discussion should address the implications of these limitations for the strength and generalizability of the conclusions.
Answer: Thank you for the critical review and expert comment, we added in the discussion & research Gap about the limitations of our study, that the heterogeneity of end points in all the included seven studies is a limitation of our study to compare the present evidence with different measured endpoints, and inability to unify the evidence and establish a numerical cut-off value, which also denotes results are not generalizable and further large scale studies are needed. (The following changes have been incorporated in the edited manuscript on page 13-14, from lines 415-452)
4. It would also be beneficial to more clearly delineate what is novel in this review relative to previous work on endocan in cardiovascular disease. In particular, emphasize the specific focus on STEMI treated with PCI, microvascular dysfunction and no-reflow, and the interactions with diabetes or stress hyperglycemia. Some sections, especially the pathophysiological discussion and the comparison with other biomarkers, could be tightened to foreground the mechanisms and aspects most relevant to STEMI and to highlight the potential incremental value of endocan beyond established markers.
Answer: Thank you for your expert comment. Yes, we have included in the discussion and conclusion section of the review article delineating the novelty our study establishes, combing all present evidence available and suggesting dynamic obstruction and microvascular dysfunction being the culprit post-PCI with dismal prognosis even with restored coronary flow and its association with Endocan levels. Also, we discussed emphasized Endocan levels with all the phenomenon and markers suggesting endothelial dysfunction. We added relevance of endocan as a marker as compared to other biomarkers presently studied. Also highlighting need for a study for establishing cut-off value with incremental values denoting worse prognosis. (The following changes have been incorporated in the edited manuscript on page 13-14, from lines 415-452)
- Discussion & Research Gap
A better understanding of endocan with respect to other inflammatory conditions like sepsis, CKD, IBD, and tumor progression can help us narrow down endocan’s relationship to cardiovascular diseases; however, these associations remain exploratory and should be interpreted cautiously. It is a novel biomarker that has the potential to be used to further understand the long-term prognosis of ACS patients as well as understand the severity of the disease upon admission. However, future prospective studies with larger sample sizes are required to clarify its prognostic role.
Our study delineates a concise analytical review of the present evidences available on the research literature to establish Endocan levels as a prognostic indicator particularly for patients with acute conditions like STEMI, undergoing PCI. In an era of endovascular coronary interventions, it is necessary to establish prognosis for endovascular dysfunction prior to every PCI, since it is important to explore why all endovascular coronary interventions vary in prognosis. This is especially important given present prognostic indicators do not account for post-PCI endovascular dysfunction which denote important aspect of post procedure prognosis even when anatomical obstruction is corrected. Often, patients presenting to the Cath lab with acute condition like STEMI and needing coronary interventions have multiple comorbidities like uncontrolled diabetes/ uncontrolled hyperglycemia, deranged kidney function and chronic inflammatory disorders like chronic infections, etc. Post-procedure outcome and recurrent MACE events are still common in these patients hinting dynamic flow obstruction with pathophysiological microvascular dysfunction.
Hassanpour et al, suggested Endocan levels correlated with Apo-B, which itself is an established marker of endothelial dysfunction. However, Endocan levels are more specific to ischemic and acute level of microvascular dysfunction. Similarly, galectin which is similar to Endocan levels is a marker of Endovascular dysfunction which needs to be studied more. Adding to the evidence, physiological phenomenon which predict endothelial and microvascular dysfunction post-PCI like No-reflow phenomenon and incomplete-STR correlate to Endocan levels.
However, important limitation of our study is the heterogeneity of the outcome variables in noted seven studies. Given the variable and heterogenous primary and secondary endpoints measurements in our included studies, included MACE events, physiological phenomenon like NRT or incomplete-STR, markers like Apo-B and Apo-A levels, hs-CRP. Heterogenous variables measured could not be combined to have statistically relevant evidence, that could not be measured and evidence cannot have a generalizability. Also of note, different studies denote different cut of points for Endocan levels, constraining to have particular value with established relevance. Additionally, variations in study design, follow-up duration, endpoints, and measurement techniques represent potential sources of bias and heterogeneity, which should be considered when interpreting these findings.
Further research highlighting Endocan’s correlation and establishing numerical cut-off value for prognosis is needed with bigger sample sizes. Future research can also help us incorporate Apo-B levels along with endocan into understanding the severity of cardiovascular disease. High cholesterol, alcohol, obesity, and smoking can also increase endocan levels, and future studies with stratification can help us remove the confounders and truly understand the nature of endocan levels in ACS patients. Stratification would also help us understand how much cholesterol, obesity, alcohol, and smoking really affect the Endocan levels and provide further insight into those conditions as well and their relationship with Endocan. This could help us establish a cut-off value of Endocan for each inflammatory condition. This can help us specifically target ACS and the relation of ESM-1 levels to STEMI patients. It would also be interesting to understand how endocan levels vary in other ACS conditions such as NSTEMI and unstable angina. In addition, high-risk ECG phenotypes such as Wellens syndrome have been associated with critical coronary pathology and microvascular dysfunction despite transient symptoms. Case-based reports suggest that endothelial injury and altered coronary flow dynamics may contribute to this presentation. Given endocan’s association with endothelial activation, its potential role in refined risk stratification in such phenotypes warrants further investigation. Understanding its role in angiogenesis can help us understand further how the myocyte repair correlates to prognosis and long-term outcomes. There is conflicting data as angiogenesis and myocyte repair are done by endocan at the same time it positively correlates to MACEs post-PCI. Further understanding this aspect can help us understand the nature of Endocan and its balance between favorable outcomes and adverse events post- PCI.
Finally, the manuscript would benefit from careful English language editing to correct minor errors, ensure that all abbreviations are defined on first use, standardize terminology (notably regarding STEMI wording), and harmonize reference formatting. Overall, this is a relevant and promising review. Addressing these points would enhance its clarity, rigor, and impact.
Answer: Thank you for the expert review, we corrected language and fixed minor errors/ abbreviations/ terminology/ references.

Reviewer 2 Report
Comments and Suggestions for Authors
This narrative review addresses the role of endocan/ESM-1 as a biomarker of endothelial dysfunction and prognosis in ST-elevation myocardial infarction (STEMI), integrating mechanistic insights with emerging clinical data. The authors summarize the molecular physiology of endocan, its links with inflammation, angiogenesis, and ischemia–reperfusion injury, and provide a clinically oriented discussion of no-reflow, MACE, and traditional risk scores. The schematic figures and the summary table of seven clinical studies are useful for readers and highlight the potential of endocan as part of a multimarker strategy in STEMI.
However, the manuscript remains essentially a narrative overview. The search strategy and study selection are only briefly described, and there is no formal assessment of study quality, risk of bias, or heterogeneity of outcomes and cut-off values. A more critical appraisal of the small, mostly single-centre cohorts and short follow-up is needed to temper the conclusions. At several points the text is repetitive across Introduction, Pathophysiology, Clinical Evidence, and Summary/Research Gap sections, and the balance between descriptive content and critical discussion could be improved.
Language and style require careful editing for grammar, consistency of terminology (e.g., NRP vs NRT, STR vs ST resolution), and occasional misused terms (e.g., “endocannabinoids”). Some statements, particularly regarding potential screening applications in stable coronary disease or non-cardiac inflammatory conditions, are speculative and should be clearly labelled as such or shortened to maintain focus on STEMI and PCI.
I have a few suggestions:
-
Please expand the Methods section for the literature search (databases, time limits, detailed inclusion/exclusion criteria) and explain why only seven studies were retained.
-
Consider adding at least a brief assessment of study quality and major sources of bias/heterogeneity (design, endpoints, follow-up, cut-off definitions).
-
Reduce repetition between Sections 1, 2, 3, 5 and 6; some paragraphs on pathophysiology and clinical implications could be shortened and merged.
-
Standardise terminology and abbreviations throughout (NRP vs NRT, STR vs ST-segment resolution, “no-reflow” vs “no-reflow phenomenon”) and correct occasional wording errors (e.g., “endocannabinoids”).
-
Table 2 would be more informative if it included follow-up duration, primary endpoints, and whether endocan retained independent prognostic value in multivariable models.
-
Please moderate or clearly label as speculative the proposed uses of endocan for screening in stable CAD or non-cardiac inflammatory conditions, and keep the main message centred on STEMI and PCI.
-
You may wish to briefly discuss the potential role of endocan in specific high-risk ECG phenotypes such as Wellens syndrome, where endothelial dysfunction and microvascular impairment may be particularly relevant. For instance, the reports “A Particular Case of Wellens’ Syndrome” (Medical Hypotheses 2020;144:110013, DOI: 10.1016/j.mehy.2020.110013) and “Myocardial bridging – an unusual cause of Wellens syndrome: A case report” (Medicine 2020;99(41):e22491, DOI: 10.1097/MD.0000000000022491) could be cited to illustrate how endocan might contribute to refined risk stratification in these scenarios.
Overall, this is a timely and potentially useful overview, but it would benefit from tighter structure, clearer methodological framing, and a more cautious, critical interpretation of the existing evidence base.
Author Response
Reviewer 2:
Comments and Suggestions for Authors
Things highlighted in red in the manuscript indicate the changes made in response to reviewer comments
Comment 1: Please expand the Methods section for the literature search (databases, time limits, detailed inclusion/exclusion criteria) and explain why only seven studies were retained.
Response1:
Thank you for your insightful comment. We have addressed your suggestion by adding details on the databases searched, the search timeline, the inclusion and exclusion criteria, and the rationale for retaining seven studies, which have now been incorporated into the Methods section of our manuscript. (The following changes have been incorporated in the edited manuscript on page 8, from lines 281-322)
4.2. B. Study Design and Limitations
Our literature search included a meticulous search through PubMed, Scopus, Embase, Cochrane Registry and Google scholar research databases. The studies included were recent to 1st January 2015 and searched through end of 28th February 2025. The keywords were included “Endocan,” “ESM-1,” “ST-elevated myocardial infarction (STEMI),” “acute coronary syndrome (ACS),” “reperfusion injury,” “TIMI score,” “SYNTAX score,” “angiogenesis,” and “endothelial dysfunction.”
Inclusion criteria included were observational / cross-sectional studies measuring patients admitted to hospital with ACS and STEMI along with measuring Endocan levels and correlated prognostic patient outcomes. Only studies with sample size at least >10 and published after 1st January 2015 were included.
Exclusion criteria were studies with sample size<10, studies older than 1st January 2015. Through the database, review articles, case reports, case series, letter to editor, preprint, clinical trial and experimental studies, if found were also excluded.
After a careful manual review of two researchers independently working (S.K. and P.S.), initial 10 articles were screened-in and then 3 articles were screened out following predefined search criteria and quality assessment methods. Disagreements between these 2 authors were resolved by consensus, including a third author (R.D.). The exact terms applied in each database from this search strategy are shown in the Prisma diagram as below.
After a meticulous review, a total of seven studies met inclusion criteria and a combined sample size of 1,011 patients (ranging from 80 to 320 per study). It is to be noted that, all studies included did not have a uniform outcome/ prognosis assessment. All studies evaluated ACS patients with STEMI undergoing PCI, assessing correlations between endocan levels and outcomes such as no-reflow phenomenon, TIMI score, hs-CRP, SYNTAX score, type 2 diabetes, and incomplete-ST resolution.
Most studies utilized control cohorts from the same population, ensuring comparability and strengthening internal validity. Uniform data collection across single-center cohorts further enhanced methodological consistency. However, potential confounders such as smoking, alcohol use, hypercholesterolemia, obesity, and postmenopausal status may influence endocan levels independently, limiting external generalizability. Nonetheless, the findings support endocan’s potential as a biomarker for both angiogenesis and prognostic assessment following STEMI treated with PCI.
Comment 2: Consider adding at least a brief assessment of study quality and major sources of bias/heterogeneity (design, endpoints, follow-up, cut-off definitions).
Response 2: Thank you for your valuable comment. We have added a discussion on the heterogeneity and quality of the included studies, highlighting differences in study design, endpoints, follow-up, and cut-off definitions, as well as potential sources of bias such as comorbidities and small sample sizes. These points are now incorporated in the manuscript in the section discussing study limitations and heterogeneity. (The following changes have been incorporated in the edited manuscript on page 14, from lines 438-448)
Discussion & Research Gap
A better understanding of endocan with respect to other inflammatory conditions like sepsis, CKD, IBD, and tumor progression can help us narrow down endocan’s relationship to cardiovascular diseases. It is a novel biomarker that has the potential to be used to further understand the long-term prognosis of ACS patients as well as understand the severity of the disease upon admission.
Our study delineates a concise analytical review of the present evidences available on the research literature to establish Endocan levels as a prognostic indicator particularly for patients with acute conditions like STEMI, undergoing PCI.
In an era of endovascular coronary interventions, it is necessary to establish prognosis for endovascular dysfunction prior to every PCI, since it is important to explore why all endovascular coronary interventions vary in prognosis. This is especially important given present prognostic indicators do not account for post-PCI endovascular dysfunction which denote important aspect of post procedure prognosis even when anatomical obstruction is corrected. Often, patients presenting to the Cath lab with acute condition like STEMI and needing coronary interventions have multiple comorbidities like uncontrolled diabetes/ uncontrolled hyperglycemia, deranged kidney function and chronic inflammatory disorders like chronic infections, etc. Post-procedure outcome and recurrent MACE events are still common in these patients hinting dynamic flow obstruction with pathophysiological microvascular dysfunction.
Hassanpour et al, suggested Endocan levels correlated with Apo-B, which itself is an established marker of endothelial dysfunction. However, Endocan levels are more specific to ischemic and acute level of microvascular dysfunction. Similarly, galectin which is similar to Endocan levels is a marker of Endovascular dysfunction which needs to be studied more. Adding to the evidence, physiological phenomenon which predict endothelial and microvascular dysfunction post-PCI like No-reflow phenomenon and incomplete-STR correlate to Endocan levels.
However, important limitation of our study is the heterogeneity of the outcome variables in noted seven studies. Given the variable and heterogenous primary and secondary endpoints measurements in our included studies, included MACE events, physiological phenomenon like NRT or incomplete-STR, markers like Apo-B and Apo-A levels, hs-CRP. Heterogenous variables measured could not be combined to have statistically relevant evidence, that could not be measured and evidence cannot have a generalizability. Also of note, different studies denote different cut of points for Endocan levels, constraining to have particular value with established relevance. Additionally, variations in study design, follow-up duration, endpoints, and measurement techniques represent potential sources of bias and heterogeneity, which should be considered when interpreting these findings.
Further research highlighting Endocan’s correlation and establishing numerical cut-off value for prognosis is needed with bigger sample sizes. Future research can also help us incorporate Apo-B levels along with endocan into understanding the severity of cardiovascular disease. High cholesterol, alcohol, obesity, and smoking can also increase endocan levels, and future studies with stratification can help us remove the confounders and truly understand the nature of endocan levels in ACS patients. Stratification would also help us understand how much cholesterol, obesity, alcohol, and smoking really affect the Endocan levels and provide further insight into those conditions as well and their relationship with Endocan. This could help us establish a cut-off value of Endocan for each inflammatory condition. This can help us specifically target ACS and the relation of ESM-1 levels to STEMI patients. It would also be interesting to understand how endocan levels vary in other ACS conditions such as NSTEMI and unstable angina. It is not known if there is a linear correlation between angina, unstable angina, NSTEMI, and STEMI, but it would potentially be helpful in preventing a STEMI in case there is a relationship by screening the patients with Endocan before they have symptoms of angina. Understanding its role in angiogenesis can help us understand further how the myocyte repair correlates to prognosis and long-term outcomes. There is conflicting data as angiogenesis and myocyte repair are done by endocan at the same time it positively correlates to MACEs post-PCI. Further understanding this aspect can help us understand the nature of Endocan and its balance between favorable outcomes and adverse events post- PCI.
.
Comment 3: Reduce repetition between Sections 1, 2, 3, 5 and 6; some paragraphs on pathophysiology and clinical implications could be shortened and merged.
Response 3: Thank you for your insightful comment. We have addressed your suggestion and incorporated the relevant changes into our manuscript. To improve clarity and reduce repetition, we have shortened overlapping content across Sections 1, 2, 3, 5, and 6.
Specific revisions made:
Section 1.3. Epidemiology and Clinical Impact of Endocan Over STEMI
Coronary artery disease (CAD) accounts for over 7.4 million deaths and cases of disability worldwide mainly attributed to atherosclerosis. Atherosclerosis plays a significant role in the inflammatory process, causing endothelial dysfunction leading to acute coronary syndrome (ACS), and STEMI is the main cause of acute myocardial infarction.1 Cardiovascular disease is a common complication in patients with diabetes, and studies have shown an association between stress hyperglycemia and STEMI due to endothelial dysfunction.6 According to a study, around 3-71% of patients with Acute Myocardial Infarction (AMI) are affected by stress hyperglycemia.6
ST-segment elevation myocardial infarction (STEMI) requires urgent attention to reperfusion in order to avoid complications as it can cause thrombosis in an epicardial coronary artery through atherosclerosis. Hence, Primary percutaneous coronary intervention (PCI) has become is the most effective treatment strategy for STEMI.4
Although primary PCI restores coronary blood flow in most patients with TIMI-3, some still exhibit poor restoration and in spite of vessel patency. Therefore, assessing the inflammatory markers such as endocan, which play an important role in endothelial function, remains clinically relevant.2 is crucial to understand dynamic circulation mechanisms involved at cellular level.2 Therefore, addressing patients with STEMI in whom primary PCI is unsuccessful remains a major challenge and should be further studied.
What was removed:
- Reference to unsuccessful PCI and no-reflow (already discussed in detail in Sections 2.4 and 3)
Section 2.4. D. Interplay with Ischemia-Reperfusion Injury
…
The mortality rate in STEMI and the risk of congestive heart failure, malignant arrhythmia, and sudden cardiac death significantly increase in the patients with the NRP after post-PCI. Endocan was found to be an independent predictor of NRP with a value of >2.7 ng/mL, having 89.6% sensitivity and 74.2% specificity for the prediction of the NRP. This suggests that endocan levels may be helpful in recognizing patients with a higher risk of insufficient myocardial perfusions and worse clinical outcomes post PCI.2
Supporting this, a study done by Turan et al. showed that elevated endocan (and galectin-3) levels may help identify patients at risk of poor myocardial perfusion and adverse outcomes after post-PCI.4 Moreover, a high endocan level on hospital admission might serve as an independent predictor of worse cardiovascular outcomes and higher TIMI risk scores in patients with ACS.3 …
What was removed
- Predictive statistics and outcome associations - already covered in Section 3 (evidence) and Section 4.3 (clinical utility)
Section 4.3. C. Endocan as a Prognostic Tool in Patients with STEMI
…
This study also suggested the possibility of potential lack of reperfusion causing an increase in Endocan levels, further supporting the study by Dogdus et al.2 Additionally, in the study by Qui et al6, a multiple factors logistic regression analysis in the study indicated that an ESM-1 level >1.01 ng/mL was an independent predictor of MACE. This further supports that Endocan levels can be used as a predictor to understand MACE.
but further research is required in this field.
It is also not clear how the endocan levels play out through the course of an admission for STEMI, including before and after intervention. Hence, further prospective, multicenter studies to understand the Endocan levels before intervention and post-intervention would be useful to understand the trend of Endocan levels throughout the course of an admission in patients with STEMI. It would be useful to see how the endocan levels play out throughout the course before STEMI and the time of intervention, focusing on survival, mortality, and long-term outcomes, including MACE, quality of life, and recurrent admissions.
What was removed:
- future-study repetition, mentioned in Section 6
- Endocan trends in MI, mentioned in section 5
Section 5. summary and clinical implications (The following changes have been incorporated in the edited manuscript on page 13, from lines 396-397 ; 402-405)
Endocan levels have the potential to be a good screening tool for STEMI, as suggested by the linear correlation between endocan and TIMI score. It can further be incorporated in routine workup for ACS as an inflammatory marker; however, its clinical utility may be limited by confounding inflammatory conditions such as sepsis.44 Further research to mark a cut-off endocan level in ACS and, more specifically, STEMI can be useful. ESM-1 levels can also be used for understanding the outcomes post-PCI, including MACE as suggested by Qui et al6.
It would be useful to see how the endocan levels play out throughout the course before STEMI and the time of intervention, focusing on survival, mortality, and long-term outcomes, including MACE, quality of life, and recurrent admissions.
Other factors such as survival, mortality, and quality of life can also be studied and have the potential to be correlated to endocan post-intervention. Additionally, endocan can also be useful to understand the angiogenesis following a STEMI and to predict NRP.
It would be interesting to see how it correlates, and a prospective multicenter study would be beneficial. as ESM-1 is also proven to be a neo-angiogenesis factor and could be elevated in response to angiogenesis following MI. Understanding of the angiogenesis in MI and its relation to endocan levels can also help us predict the levels of endocan post-PCI related to NRP.
Changes made:
- To tighten sentence structure.
What was removed:
- Mention about larger sample sizes and prospective multicenter studies already covered in section 6 (research gap)
- Repetition of angiogenesis mechanisms already detailed in Section 2.3
Section 6. Research Gap (The following changes have been incorporated in the edited manuscript on page 13, from lines 412-413; page 14, from lines 461-467)
A better understanding of endocan with respect to other inflammatory conditions like sepsis, CKD, IBD, and tumor progression can help us narrow down endocan’s relationship to cardiovascular diseases. It is a novel biomarker that has the potential to be used to further understand the long-term prognosis of ACS patients as well as understand the severity of the disease upon admission. However, future prospective studies with larger sample sizes are required to clarify its prognostic role. Future research can also help us incorporate Apo-B levels along with endocan into understanding the severity of cardiovascular disease. High cholesterol, alcohol, obesity, and smoking can also increase endocan levels, and future studies with stratification can help us remove the confounders and truly understand the nature of endocan levels in ACS patients. Stratification would also help us understand how much cholesterol, obesity, alcohol, and smoking really affect the Endocan levels and provide further insight into those conditions as well and their relationship with Endocan. This could help us establish a cut-off value of Endocan for each inflammatory condition. This can help us specifically target ACS and the relation of ESM-1 levels to STEMI patients.
It would also be interesting to understand how endocan levels vary in other ACS conditions such as NSTEMI and unstable angina. In addition, high-risk ECG phenotypes such as Wellens syndrome have been associated with critical coronary pathology and microvascular dysfunction despite transient symptoms. Case-based reports suggest that endothelial injury and altered coronary flow dynamics may contribute to this presentation. Given endocan’s association with endothelial activation, its potential role in refined risk stratification in such phenotypes warrants further investigation.
It is not known if there is a linear correlation between angina, unstable angina, NSTEMI, and STEMI, but it would potentially be helpful in preventing a STEMI in case there is a relationship by screening the patients with Endocan before they have symptoms of angina.
Understanding its role in angiogenesis can help us understand further how the myocyte repair correlates to prognosis and long-term outcomes. There is conflicting data as angiogenesis and myocyte repair are done by endocan at the same time it positively correlates to MACEs post-PCI. Further understanding this aspect can help us understand the nature of Endocan and its balance between favorable outcomes and adverse events post-PCI.
Changes made:
- Tightened sentence structure
- Removed repetition
Section 7. Conclusions (The following changes have been incorporated in the edited manuscript on page 15, from lines 481-491)
…
As per our review, the diagnostic ability of endocan levels is still questioned; however, there is significant evidence beyond just mere consideration that endocan carries significant value in the prognostication of ACS patients with STEMI and can predict worse prognosis in terms of a variety of endpoints, including MACEs, NRP post-PCI, and non-STR. Future studies should be conducted with long-term follow-up to establish a diagnostic and prognostic application and cut-off value for Endocan in ACS patients with STEMI.
Changes made
Tightened sentence structure
Comment 4. Standardize terminology and abbreviations throughout (NRP vs NRT, STR vs ST-segment resolution, “no-reflow” vs “no-reflow phenomenon”) and correct occasional wording errors (e.g., “endocannabinoids”).
Response 4: We appreciate the reviewer’s careful attention to terminology and language consistency. We have thoroughly revised the manuscript to standardize abbreviations and correct wording errors.
Specific revisions made:
Abstract (The following changes have been incorporated in the edited manuscript on page 1, from lines 37)
...
Seven clinical studies, comprising sample sizes ranging from 80 to 320 patients, met the inclusion criteria. Across these studies, endocan levels were analyzed in relation to established prognostic markers and clinical outcomes. Key findings demonstrated that higher endocan levels correlated with stress hyperglycemia (r = 0.21, p < 0.05), higher SYNTAX scores, and worse in-hospital outcomes. A cutoff value of 1.7 ng/mL predicted STEMI with 76.1% sensitivity and 73.6% specificity. Elevated endocan levels also showed positive correlations with the TIMI risk score, major adverse cardiovascular events (MACE), and were identified as independent predictors of incomplete ST-segment resolution (STR) (p = 0.044) and no-reflow phenomenon (NRP) (p < 0.001, OR = 2.39, 95% CI = 1.37–4.15).
Section 2.2. B. Inflammation
…
STEMI is often precipitated by the rupture of thrombotic plaques. Following plaque rupture, neutrophils and monocytes are recruited to the site of injury, where they contribute to further inflammation and myocardial damage. The neutrophil-to-lymphocyte ratio (NLR) has been studied as a potential prognostic marker in STEMI, due to its reflection of systemic inflammation.26 Inflammatory markers IL-1β and IL-6 have also been associated with increased Thrombolysis in Myocardial Infarction (TIMI) risk score in STEMI, highlighting their role in risk stratification.23 Additionally, IL-6 and TNF-α have been correlated to atherosclerotic plaque destabilization.27
So, how does this help us bridge the gap between the relation of endocan to inflammation and further to STEMI?
Section 3. Clinical Evidence Linking Endocan to STEMI Outcomes(The following changes have been incorporated in the edited manuscript on page 7, from lines 261)
Current clinical evidence suggests Endocan as a biomarker for endothelial dysfunction and ischemic conditions along with a positive correlation with other biomarkers like apolipoprotein-B (APO-B).1 Endocan along with galectin-3 can predict a higher risk of insufficient myocardial reperfusion and worse clinical outcomes after post-PCI.4, 38 Studies also suggested a positive and independent correlation of endocan levels with MACE, indicating a poor prognosis with elevated levels.3, 5
“No-reflow phenomenon (NRP)” after post-PCI is an important predictor and risk factor for worse clinical outcomes after STEMI and PCI associated with MACE, heart failure, arrhythmia, and risk of sudden cardiac death.
…
Kundi et al. suggested a direct correlation of endocan with the SYNTAX score, which is a semi-quantitative measurement of the complexity of coronary artery disease, and a value of endocan >1.7 ng/ml suggested the presence of STEMI with a sensitivity of 76.1% and specificity of 73.6%5, and an endocan value of >2.7 ng/mL has 89.6% sensitivity and 74.2% specificity for the prediction of the NRP.2
Section 4.3. C. Endocan as a Prognostic Tool in Patients with STEMI (The following changes have been incorporated in the edited manuscript on page 9, from lines 343)
…
Additionally, as per the study by Turan et al4, there was also a significant correlation between endocan levels and ST segment resolution (STR) following PCI. Their hypothesis was that elevated levels of endocan, as an indicator of endothelial dysfunction and inflammation, may be surrogate markers of insufficient microvascular perfusion at the myocyte level post-PCI, which might contribute to an incomplete STR.
Section 5. Summary & Clinical Implications (The following changes have been incorporated in the edited manuscript on page 13, from lines 402)
Other factors such as survival, mortality, and quality of life can also be studied and have the potential to be correlated to endocan post-intervention. Additionally, endocan can also be useful to understand the angiogenesis following a STEMI and to predict NRP.
Section 7. Conclusions (The following changes have been incorporated in the edited manuscript on page 15, from lines 489-491)
…
As per our review, the diagnostic ability of endocan levels is still questioned; however, there is significant evidence beyond just mere consideration that endocan carries significant value in the prognostication of ACS patients with STEMI and can predict worse prognosis in terms of a variety of endpoints, including MACEs, NRP post-PCI, and non-STR. Future studies should be conducted with long-term follow-up to establish a diagnostic and prognostic application and cut-off value for Endocan in ACS patients with STEMI.
Table 1
Endocan (last column) - Elevated endocan in STEMI patients with stress hyperglycemia is associated with a higher risk of MACEs [12]
Endocan levels may help identify patients at risk of poor myocardial perfusion and the “NRP”, and adverse outcomes after PCI.[2],[4]
Changes made:
- NRT → NRP
- ST resolution → ST-segment resolution (STR)
- “no reflow” → no-reflow phenomenon (NRP)
- Endocannabinoids → endocan
Comment 5: Table 2 would be more informative if it included follow-up duration, primary endpoints, and whether endocan retained independent prognostic value in multivariable models. (The following changes have been incorporated in the edited manuscript on page 11-13, from lines 391-392)
Response 5: Thank you for this helpful suggestion. We have updated Table 2 to include follow-up duration, primary endpoints, and whether endocan retained independent prognostic value in multivariable analyses. The revised table has been incorporated into the manuscript
|
|
|
|
Table 2. Studies with clinical evidence of Endocan association with STEMI outcome |
|||||||||
|
|
Study |
Sample size |
Study Type |
Objective |
Key Methods/Measurements |
Follow-up duration |
Primary Endpoint |
Independent prognostic value of Endocan |
Key Findings |
Conclusion |
||
|
1 |
Hassanpour et al 2024 |
80 men |
Crosssectional study |
Investigate serum endocan levels in STEMI patients and their correlation with apolipoproteins (APO-A1 and APO-B). |
Measurement of serum endocan, APO-A1, and APO-B; correlation analysis between biomarkers. |
Not applicable |
Biomarker correlation [Endocan vs APO-A1, APO-B, APO-B/APO-A1 ratio |
Not assessed [no clinical outcomes or multivariable outcome model] |
STEMI cases showed significant differences in APO-A1, APO-B, endocan, and the APO-B/APO-A1 ratio compared to controls; a significant positive correlation was found between endocan and APO-B. |
High endocan level is an independent indicator of endothelial dysfunction and ischemic conditions, potentially related to APO-B. |
||
|
2 |
Dogdus et al 2021 |
137 STEMI patients |
Prospective observational study |
Evaluate whether serum endocan levels can predict the angiographic no-reflow phenomenon (NRP) in STEMI patients undergoing primary PCI. |
Measurement of serum endocan; clinical and angiographic assessment; multivariate logistic regression; ROC curve analysis. |
In-hospital |
Angiographic No‑reflow phenomenon (NRP) |
Yes-independent predictor in multivariable logistic regression [OR 2.39, 95% CI 1.37-4.15] |
Endocan levels were significantly higher in the NRP (+) group; identified as an independent predictor (OR = 2.39) with a cutoff value >2.7 ng/mL (89.6% sensitivity, 74.2% specificity). |
Endocan levels may help identify patients at higher risk for insufficient myocardial perfusion and worse outcomes post-PCI. |
||
|
3 |
Turan et al 2020 |
98 STEMI patients |
Cross-sectional study |
Assess the relationship of endocan and galectin-3 levels with ST-segment resolution (STR) in STEMI patients undergoing PCI. |
Measurement of serum endocan and galectin-3; recording of SYNTAX scores; comparison between complete (≥70%) and incomplete (<70%) STR groups; regression analysis. |
In-hospital |
Incomplete ST‑segment resolution [<70%] |
Yes-independently predicted incomplete STR after adjustemnt |
Patients with incomplete STR had significantly higher endocan, galectin-3, and SYNTAX scores, along with adverse metabolic profiles and lower ejection fraction; both biomarkers independently predicted incomplete STR. |
Elevated endocan (and galectin-3) levels may help identify patients at risk of poor myocardial perfusion and adverse outcomes after PCI. |
||
|
4 |
Ziaee et al 2019 |
320 ACS patients (STEMI, NSTEMI, or UA) |
Prospective cross-sectional study |
Evaluate the prognostic value of serum endocan in relation to the TIMI risk score and its association with major adverse cardiac events (MACEs) in ACS patients. |
Measurement of serum endocan on admission; correlation with TIMI risk score and clinical outcomes; multivariate logistic regression analysis; determination of optimal cutoff values. |
In-Hospital+ 30-day follow-up |
MACEs (in‑hospital death, HF, recurrent MI) and correlation with TIMI score |
Yes- independently associated with MACE in multivariable model |
A significant positive correlation was found between endocan levels, TIMI risk score, and MACE; optimal cutoffs varied by ACS subtype; endocan was independently associated with MACE. |
A high Endocan score was a predictor of MACE and a positive correlation with that of TIMI score in ACS patients. |
||
|
5 |
Kundi et al 2017 |
133 patients (88 STEMI vs. 45 normal coronary arteries) |
Cross-sectional study |
Determine whether admission endocan level can predict in-hospital mortality and correlate with coronary severity (SYNTAX score) in STEMI patients. |
Measurement of serum endocan, hsCRP, peak troponin I, left ventricular ejection fraction, and SYNTAX score; ROC analysis to identify an optimal endocan cutoff. |
In-hospital |
Presence of STEMI & coronary severity (SYNTAX) |
Yes- independently associated with STEMI presence [not long term outcomes] |
Endocan independently correlated with STEMI presence and coronary severity; a cutoff of 1.7 ng/mL predicted STEMI with 76.1% sensitivity and 73.6% specificity; significant correlations with hs-CRP and SYNTAX score were noted. |
High admission endocan level is an independent predictor of adverse cardiovascular outcomes and a higher coronary severity index in STEMI patients. |
||
|
6 |
Qiu et al 2017 |
72 T2DM patients with STEMI and 33 control subjects (total n = 105) |
Pilot observational study |
Analyze serum endocan (ESM-1) levels in T2DM patients with STEMI and assess correlations with inflammatory markers. |
Measurement of serum ESM-1; comparisons between T2DM with vascular disease, without vascular disease, and controls; correlation analysis with hs-CRP and neutrophil-to-lymphocyte ratio. |
Not applicable |
Comparison of endocan levels with groups (T2DM STEMI vs others) |
Not assessed [no outcome-based multivariable model] |
T2DM patients with STEMI had significantly higher ESM-1 levels compared to controls and newly diagnosed T2DM without vascular disease; positive correlations with hs-CRP and neutrophil-to-lymphocyte ratio were observed. |
Serum ESM-1 may serve as a novel biomarker of endothelial dysfunction and is associated with vascular disease in T2DM patients. |
||
|
7 |
Qiu et al 2016 |
105 STEMI patients with stress hyperglycemia and 33 controls (total n = 138) |
Pilot observational study |
Assess the relationship between endocan (ESM-1) levels and stress hyperglycemia in STEMI patients, and evaluate its predictive value for MACEs. |
Measurement of serum ESM-1; correlation with admission glucose levels; multivariate logistic regression to evaluate prediction of MACEs over a 3-month follow-up. |
3 months |
MACEs at 3 months |
Yes- independently predicted MACE [OR 3.01; 95% CI 1.05-8.64] |
Patients with stress hyperglycemia had significantly higher ESM-1 levels; a positive correlation between ESM-1 and glucose levels was noted; ESM-1 levels >1.01 ng/mL independently predicted MACEs (OR = 3.01). |
Elevated ESM-1 in STEMI patients with stress hyperglycemia is associated with a higher risk of MACEs, supporting its role as an independent prognostic biomarker. |
||
Comment 6. Please moderate or clearly label as speculative the proposed uses of endocan for screening in stable CAD or non-cardiac inflammatory conditions, and keep the main message centred on STEMI and PCI.
Response 6: Thank you for the suggestion. We have moderated the text regarding potential uses of endocan in stable CAD or non-cardiac inflammatory conditions, clearly labeling these as speculative. The focus of the manuscript is now centered on STEMI and PCI, reflecting the evidence presented in the included studies. (The following changes have been incorporated in the edited manuscript on page 13, from lines 409-410)
Revision made:
Discussion and Research gap
A better understanding of endocan with respect to other inflammatory conditions such as sepsis, CKD, IBD, and tumor progression may help contextualize endocan’s relationship to cardiovascular diseases; however, these associations remain exploratory and should be interpreted cautiously
Comment 7. You may wish to briefly discuss the potential role of endocan in specific high-risk ECG phenotypes such as Wellens syndrome, where endothelial dysfunction and microvascular impairment may be particularly relevant. For instance, the reports “A Particular Case of Wellens’ Syndrome” (Medical Hypotheses 2020;144:110013, DOI: 10.1016/j.mehy.2020.110013) and “Myocardial bridging – an unusual cause of Wellens syndrome: A case report” (Medicine 2020;99(41):e22491, DOI: 10.1097/MD.0000000000022491) could be cited to illustrate how endocan might contribute to refined risk stratification in these scenarios.
Response 7: Thank you for this insightful suggestion. We have addressed this comment by adding a brief paragraph to the Research Gap section to discuss the potential relevance of endocan in high-risk ECG phenotypes such as Wellens syndrome. We have also ensured removal of speculative sentences. (The following changes have been incorporated in the edited manuscript on page 14, from lines 461-467)
The following paragraph has been added on page , from lines
Revision made:
Section 6. Discussion and Research gap
…
Stratification would also help us understand how much cholesterol, obesity, alcohol, and smoking really affect the Endocan levels and provide further insight into those conditions as well and their relationship with Endocan. This could help us establish a cut-off value of Endocan for each inflammatory condition. This can help us specifically target ACS and the relation of ESM-1 levels to STEMI patients.
It would also be interesting to understand how endocan levels vary in other ACS conditions such as NSTEMI and unstable angina. In addition, high-risk ECG phenotypes such as Wellens syndrome have been associated with critical coronary pathology and microvascular dysfunction despite transient symptoms. Case-based reports suggest that endothelial injury and altered coronary flow dynamics may contribute to this presentation. Given endocan’s association with endothelial activation, its potential role in refined risk stratification in such phenotypes warrants further investigation.
DOI: 10.1016/j.mehy.2020.110013 (Reference added)
It is not known if there is a linear correlation between angina, unstable angina, NSTEMI, and STEMI, but it would potentially be helpful in preventing a STEMI in case there is a relationship by screening the patients with Endocan before they have symptoms of angina.
Understanding its role in angiogenesis can help us understand further how the myocyte repair correlates to prognosis and long-term outcomes. There is conflicting data as angiogenesis and myocyte repair are done by endocan at the same time it positively correlates to MACEs post-PCI. Further understanding this aspect can help us understand the nature of Endocan and its balance between favorable outcomes and adverse events post- PCI.
Comment 8. Overall, this is a timely and potentially useful overview, but it would benefit from tighter structure, clearer methodological framing, and a more cautious, critical interpretation of the existing evidence base.
Response 8: Thank you for your valuable feedback. We have revised the manuscript to improve its structure, strengthen the methodological framing, and adopt a more cautious and critical interpretation of the existing evidence base, which has enhanced the overall clarity and quality of the manuscript.

Round 2
Reviewer 2 Report
Comments and Suggestions for Authors 64 I have seen the additions and I believe the article is now publishable.